# Processing Parameters and Ion Excipients Affect the Physicochemical Characteristics of the Stereocomplex-Formed Polylactide-b-Polyethylene Glycol Nanoparticles and Their Pharmacokinetics

**DOI:** 10.3390/pharmaceutics14030568

**Published:** 2022-03-04

**Authors:** Kohei Ogawa, Hidemasa Katsumi, Yasushi Moroto, Masaki Morishita, Akira Yamamoto

**Affiliations:** 1Formulation R&D Laboratory, CMC R&D Division, Shionogi Co., Ltd., Amagasaki-shi 660-0813, Japan; kohei.ogawa@shionogi.co.jp (K.O.); yasushi.moroto@shionogi.co.jp (Y.M.); 2Department of Biopharmaceutics, Kyoto Pharmaceutical University, Yamashina-ku, Kyoto 607-8414, Japan; morishita@mb.kyoto-phu.ac.jp (M.M.); yamamoto@mb.kyoto-phu.ac.jp (A.Y.)

**Keywords:** polymeric nanoparticle, anti-solvent precipitations, ion excipients, flow rate, process parameters, pharmacokinetics, characterization

## Abstract

To optimize the characteristics of stereocomplex polylactide-b-polyethylene glycol nanoparticles (SC-PEG NPs) in terms of pharmacokinetics (PK), we chose continuous anti-solvent precipitation with a T-junction as a preparation method and investigated the effect of using solvents containing an ion excipient (lithium bromide, LiBr) on the characteristics of SC-PEG NPs by changing the processing temperature and total flow rate (TFR). Processing temperatures above the melting temperature (T_m_) of the PEG domain produced a sharper polydispersity and denser surface PEG densities of SC-PEG NPs than those produced by processing temperatures below the Tm of the PEG domains. Response surface analysis revealed that a higher LiBr concentration and slower TFR resulted in larger and denser hydrodynamic diameters (D_h_) and surface PEG densities, respectively. However, a high concentration (300 mM) of LiBr resulted in a decreased drug loading content (DLC). ^14^C-tamoxifen-loaded ^111^In-SC-PEG NPs with larger D_h_ and denser surface PEG densities showed a prolonged plasma retention and low tissue distribution after intravenous injection in mice. These results indicate that the novel strategy of using solvents containing LiBr at different processing temperatures and TFR can broadly control characteristics of SC-PEG NPs, such as D_h_, surface PEG densities, and DLC, which alter the PK profiles and tissue distributions.

## 1. Introduction

Polymeric nanoparticles (NPs) have been intensively developed worldwide for various purposes, especially the treatment of cancer [1,2,3,4]. Polymeric NPs have been chemically synthesized and modified to provide various physicochemical characteristics, such as hydrophilic-lipophilic balance, targeting moiety, size, and surface properties [5,6,7]. Due to the stealth properties of polyethylene glycol (PEG), polymeric NPs for passive targeting increase blood circulation time and accumulate in tumor tissues (i.e., enhanced permeability and retention effects) [8,9]. In addition, surface-modified, or grafted, NPs were investigated for active targeting or tissue engineering. [10,11] Thus, polymeric NPs would be promising delivery systems.

Polylactide-b-polyethylene glycol (PLA-PEG) NPs are a promising polymeric NP delivery system. The PLA-PEG NPs are biodegradable and biocompatible. Some PLA-PEG NPs have been developed for the treatment of cancer and have progressed to clinical trials [3,4]. PLA-PEG NPs form via hydrophobic interaction of the PLA domain and are stabilized by the hydrophilic PEG domain. Hydrophobic drugs can be encapsulated in the PLA core and are subsequently released slowly. We recently performed orthogonal characterization studies to identify the relationship between the physicochemical properties of PLA-PEG NPs composed of different lactide isomers and to determine their pharmacokinetic (PK) profiles [12]. We also elucidated that stereocomplex PLA-PEG (SC-PEG) NPs composed of an equimolar mixture of l-lactide-based PLA-PEG (PLLA-PEG_uni_) and d-lactide-based PLA-PEG (PDLA-PEG_uni_) were the best formulations to increase blood circulation time due to their core crystallinity and high surface PEG density.

The guidelines for polymeric NPs recommend the characterization of NPs to identify the critical quality attributes, data concerning the consistency from nonclinical studies to human use, and reproducibility in the manufacture of polymeric NPs [13,14]. However, few reports have investigated the manufacture or preparation process of polymeric NPs [15,16] or have elucidated the relationship between the formulation attributes and PK profiles [17].

In the present study, we investigated the relationship between the process parameters of prepared NPs and their physicochemical properties in terms of PK. A continuous anti-solvent precipitation procedure with a T-junction was selected to prepare the SC-PEG NPs. This procedure has been used to prepare NPs [18,19] and is superior to the conventional bulk preparation method, such as the solvent evaporation method, in terms of scalability and reproducibility [20,21]. It was also reported that process parameters, such as temperature, flow rate, and flow rate ratio, affect the characteristics of NPs [21,22]. In addition, the conformations of macromolecules in a good solvent can be controlled by adding LiBr [23,24]. However, the application of LiBr in the preparation of NPs has not been described. Accordingly, a good solvent containing LiBr is expected to be a novel strategy to control the characteristics of the prepared SC-PEG NPs.

We focused on the processing temperature, which was set based on the physicochemical properties of the PLA-PEG unimer, such as the melting temperature (T_m_) and total flow rate (TFR), and investigated the effects of a good solvent containing ion excipients, such as lithium bromide (LiBr), on the physicochemical properties of SC-PEG NPs. The effects of varying the processing parameters on the characteristics of SC-PEG NPs were determined by orthogonal procedures, such as proton nuclear magnetic resonance (^1^H-NMR), size-exclusion chromatography–multiangle light scattering (SEC–MALS), and solid-phase extraction (SPE). Finally, PK studies were performed with SC-PEG NPs with different physicochemical properties, prepared using different processing parameters.

## 2. Materials and Methods

### 2.1. Chemicals and Reagents

Methoxy-polyethylene glycol (mPEG, 5kDa), deuterium chloroform, and deuterium oxide were purchased from Sigma Aldrich (St. Louis, MO, USA). l-lactide and tin (II) 2-ethylhexanoate (Sn (Oct)_2_) were purchased from Tokyo Chemical Industry (Tokyo, Japan). d-lactide was purchased from Leap Labchem (Hangzhou, China). Tamoxifen-free base (TAM) was purchased from MP Biomedicals (Santa Ana, CA, USA). Dichloromethane (DCM), diethyl ether, toluene, dimethyl sulfoxide, and N, N-dimethyl formamide (DMF, HPLC grade) were purchased from Fujifilm Wako Chemical Industries (Tokyo, Japan). Acetonitrile (MeCN, HPLC grade) and LiBr were purchased from Kanto Chemicals (Tokyo, Japan). Soluene 350 was purchased from Perkin-Elmer (Waltham, MA, USA). Clear-Sol I was purchased from Nacalai Tesque (Boston, MA, USA). ^14^C-TAM [N-methyl-^14^C] was purchased from American Radiolabeled Chemicals Inc. (St. Louis, MO, USA). ^111^Indium chloride ([^111^In] InCl_3_) was supplied by Nihon Medi-Physics Co. (Tokyo, Japan). All other chemicals were obtained commercially as reagent-grade products.

### 2.2. Animals

Six-week-old male ddY mice were purchased from Japan SLC (Shizuoka, Japan). The animals were maintained under specific pathogen-free conditions. All animal experiments were conducted according to the principles and procedures outlined in the National Institutes of Health Guide for the Care and Use of Laboratory Animals. The Animal Experimentation Committee of Kyoto Pharmaceutical University approved all experimental protocols involving animals.

### 2.3. Synthesis of l-Lactide-Based PLA-PEG (PLLA-PEG_uni_) and d-Lactide-Based PLA-PEG (PDLA-PEG_uni_)

PLLA-PEG_uni_ and PDLA-PEG_uni_ were synthesized using previously described procedures [12]. Briefly, mPEG 5 kDa was added to a round-bottom flask and vacuumed at 110 °C. l- or d-lactide was added to the flask for the synthesis. Sn (Oct)_2_ was also added as a catalyst. The resultant mixture was allowed to react at 160 °C for 12 h. The reactants were dissolved in DCM and then purified. Purified PLLA-PEG_uni_ and PDLA-PEG_uni_ were dried under vacuum.

### 2.4. Characterizations of Synthesized PLLA-PEG_uni_ and PDLA-PEG_uni_

#### 2.4.1. Number-Based Molecular Weight (Mn_uni_) and Polydispersity Index of Unimers (PdI_uni_) Determinations

The Mn_uni_ of PLLA-PEG_uni_ and PDLA-PEG_uni_ were evaluated by ^1^H-NMR spectroscopy (Appendix A). The Mn_uni_ was determined as previously reported [25]. PdI_uni_ were determined by the previously reported size-exclusion chromatography (SEC)-refractive index (RI) [12]. Calibrated standard polyethylene oxide (Agilent, San Diego, CA, USA) was used for calibrating the retention time and molecular weight. The relative molecular weight was calculated using μ7 plus (System Instruments, Kanagawa, Japan).

#### 2.4.2. Thermal Analysis of Synthesized PLLA-PEG_uni_ and PDLA-PEG_uni_

PLLA-PEG_uni_ or PDLA-PEG_uni_ (1 mg) was weighed into aluminum pans. Unimers were analyzed using differential scanning calorimetry (DSC, TA Instruments, New Castle, DE, USA). The temperature protocol involved equilibration at 10 °C for 2 min followed by a ramp-up of the temperature by 2 °C/min to 160 °C, equilibration at 160 °C for 2 min, and ramp-down of temperature by 2 °C/min to 10 °C. The lower temperature endothermic and exothermic peaks of PLLA-PEG_uni_ and PDLA-PEG_uni_ were defined as the T_m_ and the crystalline temperature (T_c_) of PEG, respectively. The endothermic and exothermic temperatures were defined as the T_m_ and T_c_ of the PLA, respectively [26].

#### 2.4.3. Conformation Analysis of Stereocomplex PLA-PEG Unimer (SC-PEG_uni_) in Good Solvents Containing Different Concentrations of LiBr

SEC–RI was performed to characterize the SC-PEG_uni_ conformations in solvents. The mobile phase was DMF containing 0, 20, 100, or 300 mM LiBr. The column oven temperature of 65 °C simulated the preparation conditions. Other HPLC conditions have been previously reported [12]. PLLA-PEG_uni_ and PDLA-PEG_uni_ (1 mg each) were dissolved in the corresponding LiBr concentration DMF for sample injections. The obtained chromatograms were subtracted from the blank baseline and normalized to the highest response. The relative molecular weight of the peak top (Rel. M_p_) and PdI_uni_ were calculated from polyethylene oxide standards using the procedure described in Section 2.4.1.

### 2.5. Preparation of SC-PEG NPs

PLLA-PEG_uni_ (50 mg), PDLA-PEG_uni_ (50 mg), and TAM (10 mg) were dissolved together in 1.0 mL of DMF containing 0, 20, 100, or 300 mM LiBr (considered the good solvent). The mixture was heated at 65 °C. Nine milliliters of distilled waters and 1.0 mL of the organic phase were fed at a TFR of 1 mL/min, 3 mL/min, and 8 mL/min using a Φ 0.5 mm T-junction mixer (YMC, Kyoto, Japan) at 65 °C. (Figure 1) The mixture was heated and mixed at 35 °C for the preparation temperature investigation. The resultant solutions were dialyzed to remove the solvent and residual LiBr with an 8-10 kDa MWCO dialysis membrane (Repligen, Waltham, MA, USA). Dialyzed solutions were filtered through a 0.45 µm polyvinylidene fluoride filter (Merck, Kenilworth, NJ, USA). Samples were stored at 2–8 °C until further use.

### 2.6. Characterization of TAM Loaded SC-PEG NPs

#### 2.6.1. Hydrodynamic Diameter (D_h_) and Polydispersity Index of SC-PEG NPs (PdI_NP_)

D_h_ and PdI_NP_ were determined by dynamic light scattering (DLS) using a Zetasizer Nano ZS (Malvern Instruments, Malvern, UK). The measurement methods have been described previously [12]. Each sample was diluted 1:50 in 10 mM phosphate buffer (pH 7.4) and measured by DLS.

#### 2.6.2. Determination of Encapsulation Efficiency (EE) and Drug Loading Content by Conventional Procedure (DLC_conv._) and SPE (DLC_SPE_)

Concentrations of TAM in NPs were determined using a previously described HPLC procedure [12]. Briefly, prepared NPs were diluted with acetonitrile (MeCN). TAM stock solution (1 mg/mL) was diluted with MeCN to prepare standard solutions of 100 to 1 µg/mL. The samples were analyzed using HPLC. The concentration of TAM was calculated using external standards.

The concentrations of SC-PEG_uni_ in the NPs were determined as previously described [12]. The prepared NPs were diluted with DMF and analyzed by SEC–RI. The concentration of SC-PEG_uni_ was calculated using external standard samples with concentrations ranging from 2 to 8 mg/mL. EE and the DLC_conv_ were calculated as
(1)$$\mathrm{EE}\left(\%\right)=C_{\mathrm{TAM}\;\mathrm{load}}\times V_{\mathrm{rec}}/\left(C_{\mathrm{TAM}\;\mathrm{feed}}\times V_{\mathrm{feed}}\right)\times 100$$
(2)$${\mathrm{DLC}}_{\mathrm{conv}.}\left(\mathrm{wt}\%\right)=C_{\mathrm{TAM}\;\mathrm{load}}/C_{\mathrm{SC}-\mathrm{PEG}}\times 100$$
where C_TAM load_ is the TAM concentration in the prepared NPs, V_rec_ is the volume of the recovered NP solution, C_TAM feed_ is fed TAM concentration to prepare the NPs, V_feed_ is the volume of solutions fed for dialysis, and C_SC-PEG_ is the weight-based concentration of SC-PEG_uni_ in the prepared NPs.

SPE was performed as previously described [12]. SPE columns were activated with MeCN and equilibrated with distilled water. The prepared NPs were then applied. The columns were washed with 20% MeCN to remove adsorbed NPs. The samples were eluted with MeCN and analyzed by HPLC to determine the TAM concentrations. DLC_SPE_ was calculated as
(3)$${\mathrm{DLC}}_{\mathrm{SPE}.}\left(\mathrm{wt}\%\right)=C_{\mathrm{TAM}\;\mathrm{SPE}}/C_{\mathrm{SC}\text{–}\mathrm{PEG}}\times 100$$
where C_TAM_SPE_ is the concentration quantified by SPE.

#### 2.6.3. Determination of Association Numbers (N_ass_), Gyration Radius (Rg), and Surface PEG Density

Association numbers and R_g_ were calculated as previously described [12]. Briefly, prepared TAM-loaded SC-PEG NPs were diluted to 5 mg/mL as SC-PEG_uni_. The prepared samples were analyzed by SEC–MALS (Wyatt Technology, Santa Barbara, CA, USA) to obtain the Berry’s plot. The molecular weight (M_NP_), R_g_, and N_ass_ of SC-PEG were calculated using the Astra software (Wyatt, US). N_ass_ was calculated as
(4)$$N_{\mathrm{ass}}=M_{\mathrm{NP}}/{\mathrm{Mn}}_{\mathrm{uni}}$$

The surface PEG densities were analyzed as previously described [12]. Briefly, the solvent of prepared NPs was exchanged with the Sephadex G50 column (PD-10 miniprep; Cytiva, Malborough, MA, USA) to deuterium oxide. An internal standard was added to the substituted samples. The calibration curve of mPEG (5 kDa) was also prepared using the same procedure. The prepared samples were analyzed by ^1^H-NMR using an AVANCE III device (Bruker, Billerica, MA, USA) operating at 400 MHz. The total amount of SC-PEG was measured by SEC–RI, as described above. Surface PEG densities and surface PEG contents were calculated as
(5)$$\mathrm{Surface}\;\mathrm{PEG}\;\mathrm{contents}\left(\%\right)=C_{\mathrm{PEG}\;\exp.}/C_{\mathrm{SC}-\mathrm{PEG}}\times \;100$$
(6)$$\Gamma =N_{\mathrm{ass}}\times \mathrm{Surface}\;\mathrm{PEG}\;\mathrm{contents}\left(\%\right)/{4\pi R}_{h}^{2}\times 100$$
where Γ is the density of the PEG molecules on the NP surface per 100 nm^2^, C_PEG_exp._ is the concentration calculated by ^1^H-NMR in deuterium oxide, and R_h_ is a half value of D_h_ measured by DLS.

#### 2.6.4. Evaluation of Morphology of Lyophilized NPs

The morphologies of lyophilized NPs were evaluated with field emission scanning electron microscopy (FE-SEM) according to a previously published method with slight modifications [27]. Lyophilized NPs were sputter-coated with a layer of osmium with the Neoc osmium coater (Meiwafosis, Japan). Prepared samples were observed with FE-SEM (JEOL JSM-IT500HR LV, Japan) and operated at 10 kV.

### 2.7. Preparation of “PEG Sparse-D_h_ Small” NP (NP_Sparse-Small_) and “PEG Dense-D_h_ Large” NP (NP_Dense-Large_) for PK Studies

^111^In-labeled SC-PEG NPs were prepared as previously described [12]. PLLA-PEG-DTPA was dissolved in DMF followed by the addition of distilled water. ^111^InCl_3_ and 10 mM citrate buffer (pH 5.5) were added. The reacted solutions were purified using PD-10 columns, and the water was evaporated. The resulting PLLA-PEG-DTPA-^111^In (^111^In-PLLA-PEG) was dissolved in DMF containing 20 mM or 100 mM LiBr. Radioactivity was measured using a 1480 Wizard™ 3 gamma counter (PerkinElmer, Waltham, MA, USA).

PLLA-PEG_uni_ and PDLA-PEG_uni_ (50 mg each) were dissolved in 0.15 mL of DMF containing 20 mM or 100 mM LiBr to prepare the NP_Sparse-Small_ NPs or NP_Dense-Large_, respectively. Ten microliters of ^111^In-PLLA-PEG with a radioactivity equivalent to 200 kBq was added. TAM (10 mg) was dissolved in 0.15 mL of DMF, and 10 µL of ^14^C-TAM having an equivalent radioactivity to 200 kBq was added. NPs were prepared as described in Section 2.5. TFRs of 3 and 8 mL/min were used to prepare the NP_Sparse-Small_ and NP_Dense-Large_, respectively. After dialysis with 9% sucrose, the samples were stored at 2–8 °C until administration.

### 2.8. Evaluations of PK Profiles of Radiolabeled PLA-PEG NPs Loaded with TAM

The ^111^In-labeled PLA-PEG NPs (^111^In-SC-PEG NPs) loaded with ^14^C-labeled TAM (^14^C-TAM) were administered by intravenous injection via the tail vein to 6-week-old ddY mice (370 kBq ^111^In-SC-PEG NPs /kg, 320 kBq ^14^C-TAM /kg). At 0.083, 0.5, 1, 3, 6, and 24 h after injection, blood was collected from the vena cava under isoflurane anesthesia, and the mice were sacrificed. The liver and spleen were removed, rinsed with saline, and blotted dry, and the total organ weight was weighed. The collected blood was centrifuged for 5 min at 2000× *g* to obtain plasma.

To measure ^111^In radioactivity, the collected organs and 100 µL of plasma were transferred to counting tubes, and the radioactivity of the samples was measured using the aforementioned 1480 Wizard™ 3 gamma counter [12,28].

^14^C-TAM radioactivity was measured as previously described [12,29] using a model LSC-6100 liquid scintillation counter (Aloka, Mitaka, Japan).

The ^111^In and ^14^C radioactivities were normalized based on the percentage of the dose/mL for plasma and on the percentage of dose/organ (g) for other tissues. The area under the curve (AUC) of ^111^In-SC-PEG NPs and ^14^C-TAM was calculated using the trapezoidal method. To estimate the total clearance (CL_tot_), half-life (T_1/2α_ and T_1/2β_), distribution volume (V_d_), and initial plasma concentrations (C_0_) of both isotopes, normalized radioactivities in the plasma were analyzed using the MULTI nonlinear least-squares program [30]. Two-compartment model analysis was performed.

### 2.9. Statistical Analyses

Statistical significance was analyzed by the Student’s *t*-test for multiple groups at a significance level of *p* < 0.05. To analyze the response surface of D_h_ and PdI_NP_, two-way ANOVA analysis was performed using Design-Expert software (Stat-Ease, Minneapolis, MN, USA) with a significance level of *p* < 0.05.

## 3. Results

### 3.1. Synthesis and Characterizations of Different PDLA-PEG_uni_ and PLLA-PEG_uni_

Table 1 presents Mn_uni_ and PdI_uni_ data of PDLA-PEG_uni_ and PLLA-PEG_uni_. The Mn_uni_ of the PLA domains was approximately 12,000, in good agreement with the theoretical values. PdI_uni_ was evaluated using SEC–RI analysis. The synthesized polymers were monomodal (Appendix A).

Figure 2A,B show the DSC plots of PDLA-PEG_uni_ and PLLA-PEG_uni_. With the temperature ramp-up from 10 °C to 160 °C, endothermic peaks of PEG melting were observed at approximately 50 °C in both copolymers, following endothermic peaks of PLA melting, which were observed at 139 °C and 153 °C, respectively (Figure 2A and Table 2). With the temperature ramp-down from 160 °C to 10 °C, exothermic peaks of PLA crystallization were observed at 97 °C and 119 °C in PDLA-PEG_uni_ and PLLA-PEG_uni_, respectively. Exothermic peaks of PEG crystallization were observed at approximately 25 °C (Figure 2B and Table 2).

### 3.2. Physicochemical Properties of NPs by Preparing at Different Temperature

SC-PEG NPs were prepared at 35 °C and 65 °C and evaluated by DLS, SEC–MALS, and ^1^H-NMR. The R_h_ of SC-PEG NPs prepared at 65 °C (NP_prep.65°C_) and 35 °C (NP_prep.35°C_) were comparable, though the PdI_nano_ of NP_prep.65°C_ were significantly lower than those of NP_prep.35°C_ (Figure 3A). Moreover, SEC–MALS revealed that NP_prep.35°C_ were more polydisperse than NP_prep.65°C_ (Figure 3B). The R_g_ value of NP_prep.35°C_ at an early retention time was larger than the R_g_ value of NP_prep.65°C_. ^1^H-NMR analysis revealed that surface PEG densities of NP_prep.65°C_ were significantly higher than those of NP_prep.35°C_ (Figure 3C). Therefore, 65 °C was selected as the processing temperature for further SC-PEG NP preparation.

### 3.3. Copolymer Conformation in Different LiBr Concentrations

An equimolar mixture of PDLA-PEG_uni_ and PLLA-PEG_uni_ dissolved in a mobile phase containing LiBr was measured by SEC–RI (Figure 4). Aggregates of unimers (retention time 6.6 min) were evident in the mobile phase that did not contain LiBr. In contrast, a mobile phase containing 20 mM LiBr displayed a monomodal molecular weight distribution of SC-PEG_uni_ (retention time 10.1 min). When the LiBr concentration was increased from 20 to 300 mM, the peak top of the copolymers shifted earlier, and all were monomodal. The relative molecular weight was calculated based on the polyethylene oxide standard (Appendix A). SC-PEG_uni_ dissolved in 300 mM LiBr and 100 mM LiBr was approximately 2.5 times and 2 times larger, respectively, than those dissolved in 20 mM LiBr. The PdI of SC-PEG_uni_ in 300 mM LiBr was 3.52, suggesting that the unimers were polydisperse.

### 3.4. Physicochemical Properties of NPs Prepared by Different LiBr Concentrations and TFR

#### 3.4.1. DLC Calculated by Conventional Method and SPE and Release Profile of TAM

Figure 5 presents data of DLC_conv._ (A) and DLC_SPE_ (B). In the case of NPs prepared with a solvent containing 300 mM LiBr, the DLC calculated by both procedures was significantly lower than those of the other processing conditions. EE displayed a similar trend (Appendix A). DLC_SPE_ was approximately half that of DLC_conv._. This means that approximately half of the TAM was adsorbed onto the surface of the NPs [12]. The TFR did not significantly affect the drug loading, but both DLC values were slightly increased by slowing the TFR.

Appendix A shows the release profiles of TAM from SC-PEG NPs, except for NPs prepared with 300 mM LiBr. All NPs released TAMs until 24 h after release was stopped until 48 h. TAM was released from the NPs again until 96 h.

#### 3.4.2. D_h_, PdI_NP_, and Surface PEG Densities of SC-PEG NPs Prepared by Different Processing Parameters

D_h_ and PdI_NP_ of SC-PEG NPs are shown in Appendix A, respectively. The D_h_ of NPs ranged from 54 to 122 nm by changing the TFR and the concentration of LiBr in solvent. All PdI_NP_ were <0.2, indicating monodispersion.

The effects of the interactions of LiBr concentrations and TFR on D_h_ and PdI_NP_ were evaluated using the response surface method. The results are presented in Figure 6A,B, respectively. Both were fitted using a quadratic model. The F-values of the models were 115 (*p* = 3.0 × 10^−17^) for D_h_ and 12.0 (*p* = 1.9 × 10^−5^) for PdI_NP_, indicating that both models were significant. The lack of fit for the F-value was not significant. The R^2^ values of the response surface were 0.91 for D_h_ and 0.53 for PdI_NP_. Both D_h_ and PdI_NP_ increased with increasing LiBr concentration and decreasing TFR.

The D_h_ and surface PEG densities correlated well (R^2^ = 0.64, Figure 6C). The surface PEG densities of SC-PEG NPs exhibited the same trend as that of D_h_ (Appendix A). Surface PEG densities ranged from 6.70 to 16.6 molecules/100 nm^2^. These results indicated that a faster TFR resulted in D_h_ and surface PEG densities of NPs that were smaller and sparser. The morphology of prepared NPs was spherical (Figure 5). The lower panel shows the morphology of NPs prepared with 100 mM LiBr/TFR 3 mL/min, which was slightly larger than that of NPs prepared with 20 mM LiBr/TFR 8 mL/min, (the upper panel).

### 3.5. Plasma PK Profile and Tissue Distributions of SC-PEG NPs with Different Surface PEG Densities and D_h_ Prepared Using Different Processing Parameters

As the surface PEG densities and D_h_ of SC-PEG NPs had a trade-off relationship, two types of SC-PEG NPs with different surface PEG densities and R_h_ were prepared to investigate mouse PK and tissue distribution. One was NP_Sparse-Small_ with surface densities and R_h_ of 7.90 molecules/100 nm^2^ and 61 nm, respectively, prepared with a solvent containing 20 mM LiBr and 8 mL/min of TFR. The other was NP_Dense-Large_ with surface densities and R_h_ of 16.6 molecules/100 nm^2^ and 105 nm, respectively, prepared with a solvent containing 100 mM LiBr and 3 mL/min of TFR.

These two types of NPs were injected into mice, and PK and tissue distributions were evaluated (Figure 7 and Table 3). NP_Dense-Large_ had higher plasma concentrations and AUCs of ^111^In-SC-PEG NPs and ^14^C-TAM compared to NP_Sparse-Small_ (Figure 7A,D and Table 3). NP_Dense-Large_ also displayed lower V_d_ and CL_tot_, and a longer T_1/2β_ of ^111^In-SC-PEG NPs and ^14^C-TAM than NP_Sparse-Small_ calculated by two-compartment models. T_1/2α_ were comparable to both NPs in terms of ^111^In-SC-PEG NPs and ^14^C-TAM. Both C_0_ values of NP_Dense-Large_ were higher than those of NP_Sparse-Small_.

Tissue distributions of ^111^In-SC-PEG NPs (Figure 7B,C) and ^14^C-TAM (Figure 7E,F) in the liver and spleen are shown. In the liver, NP_Dense-Large_ displayed significantly reduced tissue concentrations of ^111^In-SC-PEG NPs and ^14^C-TAM compared to NP_Sparse-Small_. Tissue concentrations in the spleen also showed the same trends in the liver, but they were not significant in both ^111^In-SC-PEG NPs and ^14^C-TAM.

## 4. Discussion

### 4.1. Preparation Temperature Affects PdI_NP_ and Surface PEG Densities of SC-PEG NPs

The preparation temperature was affected by the PdI_NP_ and surface PEG densities of the NPs (Figure 3). This was because the conformations of the PEG chains could be changed by temperature, as indicated by the DSC measurements (Figure 2). The PEG chain T_m_ of the synthesized PDLA-PEG_uni_ and PLLA-PEG_uni_ were 48.0 °C and 46.4 °C, respectively (Figure 2A). Below the T_m_, PEG chains would not interact with the solvent, leading to unimer aggregation. When unimers were dissolved in DMF, the solution was clear and aggregates were not observed, even at 35 °C. When NPs were prepared at 35 °C, the PEG chains could be partially embedded in the PLA core. This could be the reason why NP_prep35°C_ showed polydisperse and sparse surface PEG densities.

Previous studies have reported that bulk PEG chain conformations in di-block copolymers change depending on their crystalline temperature [31,32]. Moreover, it was also reported that PLA-PEG or PLGA-PEG NPs prepared by anti-solvent precipitation feature embedded PEG chains in the core [33,34].

### 4.2. Characteristics of SC-PEG NPs Were Controlled by LiBr Concentrations and TFR

Unimer conformations were changed by the addition of LiBr (Figure 4). Previous reports revealed that LiBr altered the conformations of macromolecules [23,24]. Our results from the evaluation of conformation are in good agreement with these reports. It was speculated that the addition of LiBr up to 20 mM resulted in dissociating the interactions between the hydrophilic domain of the unimer because previous research elucidated that LiBr shields the dipole moment between polymers [35]. In contrast, the addition of over 100 mM LiBr would lead to the salting-out effect, which resulted in associating the unimers with their hydrophobic domains. These would contribute the increase in N_ass_ in the formation of NPs. Previous research showed that the increment in LiBr in the DMF increased polymer–polymer interactions [36]. It was also reported that the size of lignin nanoparticles prepared by anti-solvent precipitations was controlled by structures of lignin in the solvent. The solution structures of lignin were changed by ion strength or pH [37]. Our observations are consistent with these reports. In addition, other reports also revealed that a fast flow rate or high Reynolds number reduces the particle size in anti-solvent precipitation preparations [38,39,40]. As the diffusion of a good solvent in the aqueous phase was faster at a fast flow rate, a smaller number of unimers were associated with decreased hydrophobic effects. Our results are in good agreement with this observation. Our preparation procedures combined the two aforementioned effects to regulate D_h_ and PdI_NP_. Higher concentrations of LiBr and slower TFR allowed more SC-PEG_uni_ to associate. The slower diffusion of solvent and more associated unimers in the solvent resulted in a larger D_h_ and wider PdI_NP_ of SC-PEG NP. However, the PdI_NP_ of all prepared NPs was <0.2, indicating that the prepared NPs were monodispersed. Due to these two processing parameters, the D_h_ of SC-PEG NP could be more widely controlled, from 53 to 122 nm, compared to using a single process parameter alone (Figure 6A,B).

Surface PEG densities were also correlated with the D_h_ of SC-PEG NPs (Figure 6C). This phenomenon was reasonable because the specific surface areas of the SC-PEG NPs increased when their D_h_ decreased. The surface PEG contents indicated that the ratio of surface-exposed PEG chains to whole PEG chains was not changed by changing the processing parameters (Appendix A). Previous studies also reported that surface PEG densities were controlled by adding hydrophobic polymers to NP systems to decrease the specific surface areas or conjugating the PEG chain to NPs to increase the surface densities [7,41]. Our present results are consistent with the previous data in terms of specific surface area governed by surface PEG densities.

Figure 8 shows a brief summary of the relationship between process parameters in the preparation of NPs. The addition of LiBr in the good solvent caused the association of the polymer, and the slower flow rate decreased the diffusion rate of the good solvent, which led to increasing N_ass_. D_h_ was increased by increasing N_ass_, and the specific surface areas of NPs were decreased, which resulted in the increase in surface PEG densities. These were the reasons the processing parameters affected the nanoparticle characteristics.

Using good solvents containing a high concentration (300 mM) of LiBr decreased the drug loading. Approximately half of the TAM was adsorbed on the surface of the NPs (Figure 5A,B). Anti-solvent precipitation procedures reportedly occur in three steps. The first step is supersaturation of the solute. The second step is nucleation. The third is the growth of particles, which kinetically stops their aggregation, resulting in the formation of NPs [34,42]. Using a good solvent containing LiBr resulted in an incremental change in the apparent molecular weight of the unimer, up to 2.5 times in the case of 300 mM LiBr (Figure 4 and Appendix A). This led to the shortened duration of supersaturation of unimers and growth of NPs. However, the duration of TAM did not change, because the solubility of TAM in 10 *v*/*v*% DMF aqueous solution containing 0 to 30 mM LiBr at 65 °C did not change (data not shown; all investigated conditions were below the detection limit of TAM concentrations). Moreover, the T_m_ of the stereocomplex domains was analyzed using lyophilization NPs. The addition of 300 mM LiBr resulted in a significantly higher T_m_ (Appendix A). This result also supports our hypothesis that a lower encapsulation TAM led to increased crystallinity of the core PLA domain. This is the reason why both DLC_conv._ and DLC_SPE_ were lower in NPs prepared with 300 mM LiBr. Approximately half of the TAM adsorbed on the surface of NPs was also supported by the release method (Appendix A). Until 24 h, surface-adsorbed TAM was released, followed by the release of encapsulated TAM from 48 h to 96 h. Previous studies also reported that PLGA-PEG NP-loaded hydrophobic drugs showed bi-phasic release profiles. In this system, surface-adsorbed drugs were predominantly released, followed by a continuous slow release [43]. Our results agree with this release pattern.

### 4.3. Different NP Preparation Conditions Altered PK Profiles and Biodistributions of SC-PEG NPs

PK profiles and biodistributions were evaluated using two types of NPs prepared by different conditions (Figure 7 and Table 3). NP_Dense-Large_ showed higher plasma AUCs of both ^111^In-SC-PEG NPs and ^14^C-TAM, and lower accumulations in the liver and spleen. These results suggest that higher PEG densities contribute to the reduction in distributions to these tissues due to the less exposed hydrophobic surface, which leads to the decreased clearance of NPs from blood. The CL_tot_ of NPs also supported this hypothesis. Previous studies also reported that PEG-dense NPs reduce the protein corona, which leads to recognition of the reticuloendothelial system (RES) and shortened duration of circulation [7,44]. Hydrodynamic diameters are also reportedly governed by specific surface area in the case of red blood cell membrane-coated NPs [41]. In contrast, the difference between NP_Sparse-Small_ and NP_Dense-Large_ was 61 nm and 105 nm, respectively; it was also reported that this difference was not critical for RES recognition [7,45]. A size-dependency of distributions of NPs to RES was reported [41,46]. This discrepancy would be raised from two factors. One was the fenestra diameter of the mouse liver, which ranged up to 150 nm [47]. In the present study, our prepared NPs were smaller than this fenestra opening; thus, they would be passed through them and avoid RES recognition. The other is the formation of a protein corona, which interacts with the surface of the NPs [7,44]. Once the protein corona is formed, the RES recognized NPs [48]. Most reports were not distinguished from the interaction between D_h_ and surface properties; thus, in our delivery systems, NP*_S_*_parse-Small_ could form a protein corona due to their hydrophobic surface properties, leading to an increase in distribution to the liver and spleen where RES existed. Detailed investigations such as the preparation of NP_Dense-Small_ and NP_Sparse-Large_ using newly synthesized PLA-PEG unimers are required. However, in the present study, we focused on the interactions of preparation procedures, characteristics of NPs, and PK profiles using single SC-PEG_uni_ to estimate the manufacture of SC-PEG NPs, which indicated that process parameters affected the characteristics of NPs and their PK profiles and biodistributions. Thus, surface PEG densities could be a potentially critical quality attribute in terms of changing biodistributions.

As mentioned above, TAM was adsorbed onto the surface of SC-PEG NPs. The surface-adsorbed TAM was predominantly released upon intravenous injection. The C_0_ of TAM loaded in both NPs was consistent with the corresponding DLC_SPE_, which also supports our hypothesis. Previous studies also reported that polymeric NPs could adsorb drugs on the surface of NPs and were immediately released in the physiological milieu [49,50]. Our results correspond to these previous reports.

## 5. Conclusions

In the present study, novel strategies to optimize the characteristics of SC-PEG NPs were investigated by changing the processing temperature, TFR, and using a good solvent containing ion excipients (LiBr). The processing temperature affected PdI_NP_ and surface PEG densities. Higher temperatures above the T_m_ of PEG produced a sharper distribution and higher surface PEG densities of SC-PEG NPs. Using solvents containing LiBr increased the apparent molecular weight, which increased the size of the NPs. The combination of good solvents containing LiBr and altering the TFR could control the surface PEG densities by changing the D_h_ of the SC-PEG NPs. However, a high concentration (300 mM) of LiBr caused a decrease in the DLC. The PK profiles and tissue distributions of the SC-PEG NPs were also altered by changing the processing conditions. SC-PEG NPs prepared using a solvent containing 100 mM LiBr and TFR of 3 mL/min resulted in higher AUCs and lower tissue distributions compared to those prepared using a solvent containing 20 mM LiBr and TFR of 8 mL/min. To the best of our knowledge, this is the first report showing that the combination of changing processing temperature, TFR, and using a solvent containing ion excipients (LiBr) is an effective strategy to regulate the characteristics of polymeric NPs. Processing parameters altered the PK profiles and tissue distributions. These findings should be helpful to make the manufacturing process of polymeric NPs robust and to control their critical quality attributes.

## Figures and Tables

**Figure 1 pharmaceutics-14-00568-f001:**
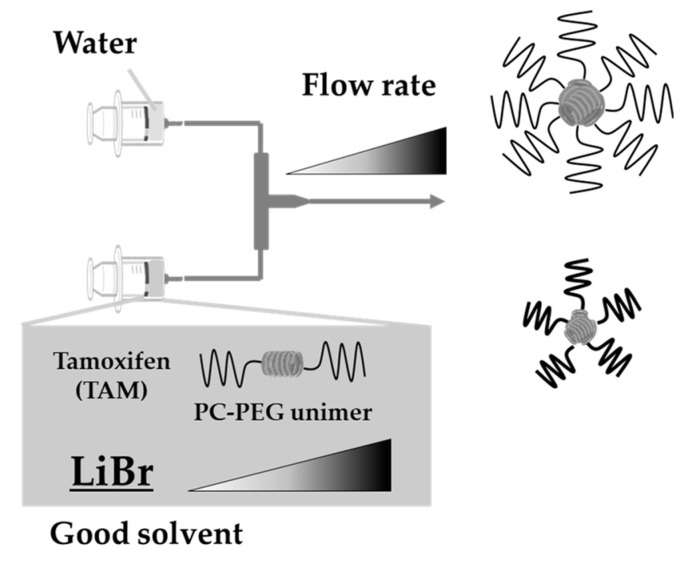
The schematic representation of the nanoparticle preparation.

**Figure 2 pharmaceutics-14-00568-f002:**
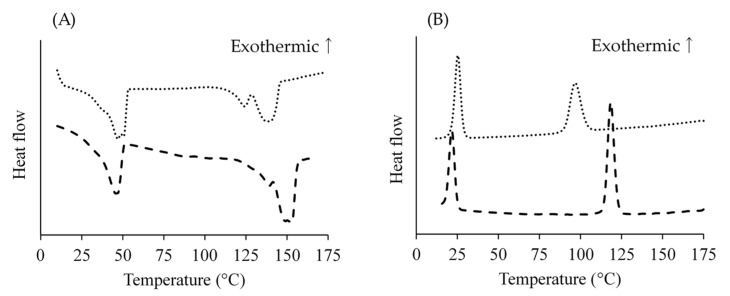
DSC plot of PDLA-PEGuni and PLLA-PEGuni. In panels A and B, the total heat flow was plotted as a function of temperature when the temperature was raised from 10 °C to 160 °C (**A**) and lowered from 160 °C to 10 °C (**B**). The dotted line and dashed line indicate heat flow of PDLA-PEGuni and PLLA-PEGuni, respectively.

**Figure 3 pharmaceutics-14-00568-f003:**
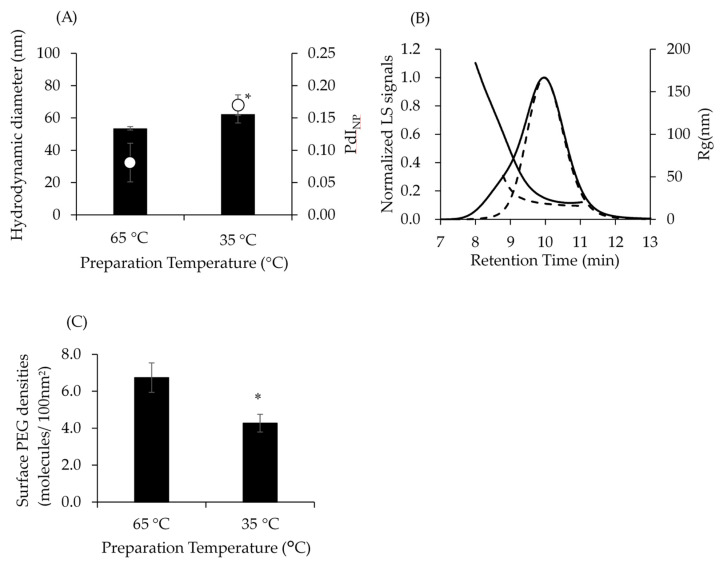
Characteristics of SC-PEG NPs prepared in different temperatures. Panel (**A**) shows the hydrodynamic diameters (D_h_) and polydispersity index (PdI_NP_). Solid bars and open circles indicate Dh (left axis) and PdI_NP_ (right axis), respectively. Results are expressed as the mean ± standard deviation of three samples. The experiments were independently performed three times; * *p* < 0.05. Panel (**B**) represents the SEC–MALS chromatogram. Solid lines and dashed lines indicate NPs prepared at 35 °C and 65 °C, respectively. Left and right axis represents light scattering signals and gyration radius (R_g_), respectively. Panel (**C**) shows surface PEG densities. Results are expressed as the mean ± standard deviation of three samples. The experiments were independently performed three times; * *p* < 0.05.

**Figure 4 pharmaceutics-14-00568-f004:**
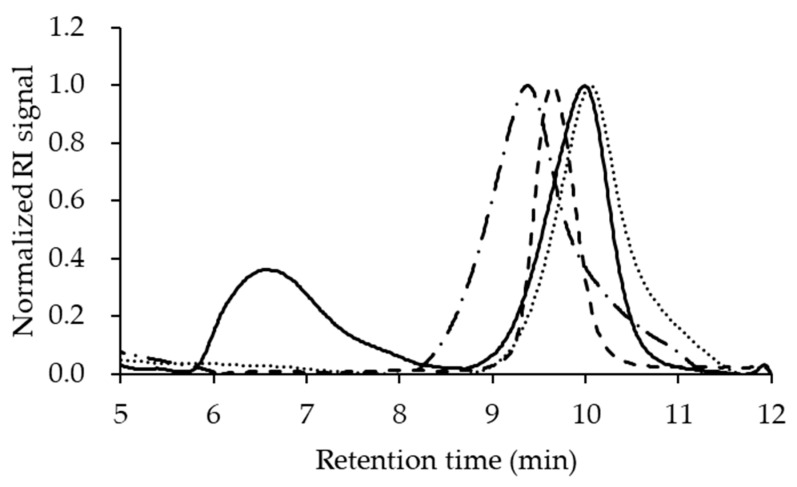
Size-exclusion chromatograms of SC-PEG_uni_ in solvents containing different concentrations of LiBr. Normalized refractive index signals were plotted as a function of retention time. The solid line, dotted line, dashed line, and chain line indicate DMF containing 0 mM, 20 mM, 100 mM, and 300 mM LiBr, respectively.

**Figure 5 pharmaceutics-14-00568-f005:**
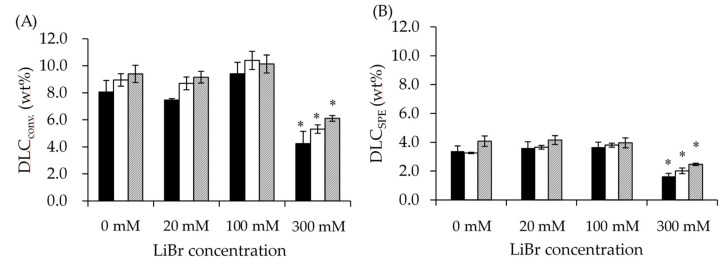
Drug loading contents (DLC) quantified by conventional method (DLC_conv._, **A**) and solid-phase extraction (DLC_SPE_, SPE) method (**B**). Filled bars, open bars, and hatched bars indicate flow rates of 8 mL/min, 3 mL/min, and 1 mL/min, respectively. Results are expressed as the mean ± standard deviation of three samples. The experiments were independently performed three times; * *p* < 0.05 to all other LiBr concentrations.

**Figure 6 pharmaceutics-14-00568-f006:**
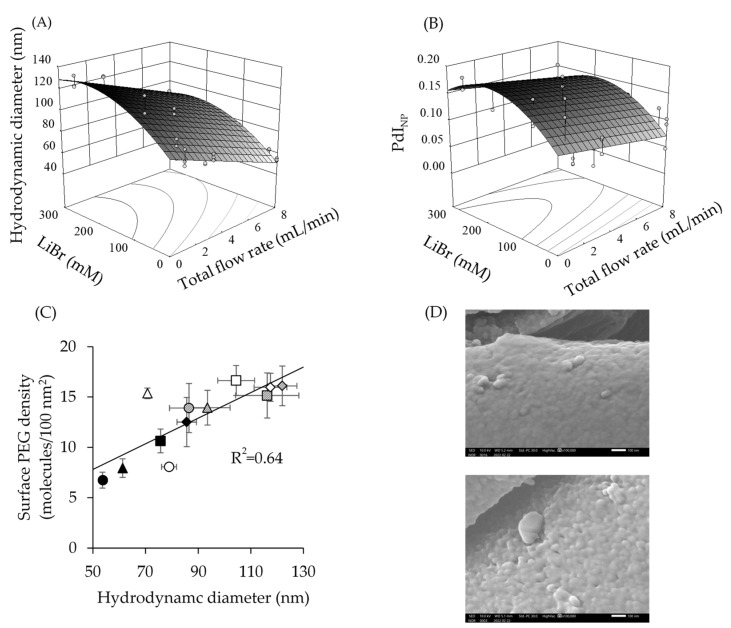
Response surface of hydrodynamic diameters (D_h_) and polydispersity (PdI_NP_) of nanoparticles (NPs), relationship between D_h_ and surface PEG densities, and FE-SEM image of lyophilized NPs. Panels (**A**,**B**) show the response surface of D_h_ and PdI_NP_, respectively. Circle symbols indicate individual values. Response surfaces were fitted by quadratic models. Panel (**C**) indicates surface PEG densities plotted as a function of D_h_. Closed, open, and hatched symbols indicate TFRs of 8 mL/min, 3 mL/min, and 1 mL/min, respectively. Circles, triangles, squares, and diamonds indicate LiBr concentrations = 0 mM, 20 mM, 100 mM, and 300 mM, respectively. Results are expressed as the mean ± standard deviation of three samples. Correlation curves were fitted by a linear model using the least squares method. Panel (**D**) shows the FE-SEM image magnified 100,000 times. The upper panels show morphologies of lyophilized NPs prepared with 20 mM LiBr/TFR 8 mL/min. The lower panel shows the morphology of lyophilized NPs prepared with 100 mM LiBr/TFR 3 mL/min. Bars represent 100 nm scales.

**Figure 7 pharmaceutics-14-00568-f007:**
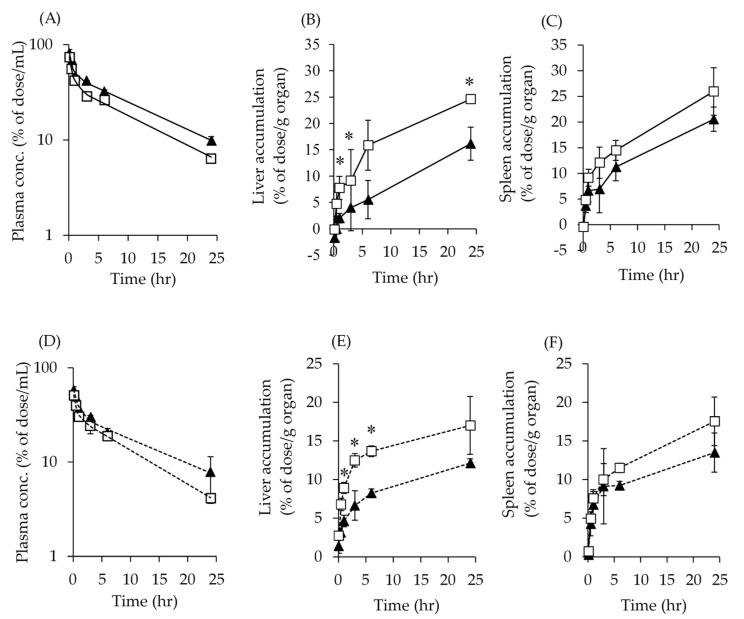
Pharmacokinetics and tissue distributions of ^14^C-TAM-loaded ^111^In-SC-PEG nanoparticles (NPs). Panels (**A**–**C**) show normalized plasma concentration, liver distributions, and spleen distributions of ^111^In-SC-PEG NPs, respectively. Panels (**D**–**F**) show normalized plasma concentration, liver distributions, and spleen distributions of ^14^C-TAM, respectively. Filled triangles and open squares indicate SC-PEG NPs prepared by LiBr 100 mM at a flow rate of 3 mL/min, and LiBr 20 mM at a flow rate of 8 mL/min, respectively. Solid lines and dashed lines in panels A and D, respectively, indicate two-compartment model fitted curves. Results are expressed as mean ± SD of three mice: * *p* < 0.05.

**Figure 8 pharmaceutics-14-00568-f008:**
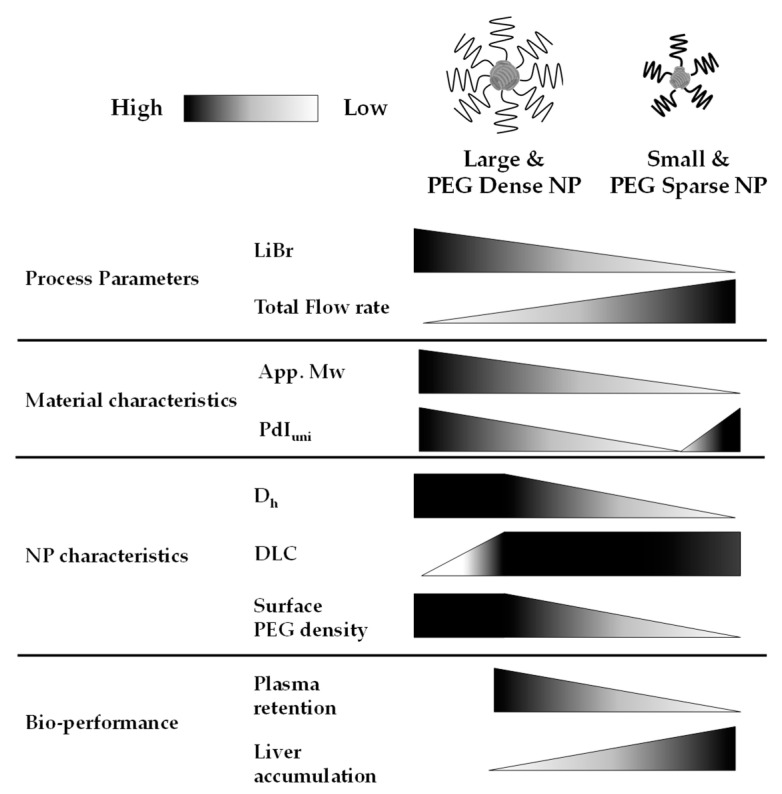
Schematic summary of the relationship between process parameters in the preparation of NPs.

**Table 1 pharmaceutics-14-00568-t001:** Number-based molecular weight and polydispersity index of synthesized PLA-PEG unimer.

Unimers	Mn PEG (g/mole)	Mn * PLA (g/mole)	PdI_uni_ **
PDLA-PEG_uni_	5000	12,007	1.78
PLLA-PEG_uni_	5000	12,497	1.82

*: Mn calculated by 1H-NMR, **: PdI calculated by SEC–RI.

**Table 2 pharmaceutics-14-00568-t002:** Melting temperature (Tm) and crystalline temperature (Tc) of synthesized PLA-PEG characterized by differential scanning calorimetry.

Unimers	PEG T_m_ ( °C)	PEG T_c_ ( °C)	PLA T_m_ ( °C)	PLA T_c_ ( °C)
PDLA-PEG_uni_	48.0 ± 2.0	25.2 ± 0.6	139 ± 0.2	96.7 ± 0.1
PLLA-PEG_uni_	46.4 ± 0.8	21.7 ± 0.1	153 ± 0.5	119 ± 0.1

**Table 3 pharmaceutics-14-00568-t003:** Pharmacokinetics parameters of PEG_Small-Sparse_ NPs and PEG_Large-Dense_ NPs.

	Plasma AUC * (h *% of dose/mL)	T _1/2 α_** (min)	T _1/2 β_ ** (min)	CL_tot_ ** (mL/min)	V_d_ ** (mL)	C_0_ ** (% of dose/mL)
^111^In-SC-PEG						
NP_Large-Dense_	640	0.374	10.5	0.133	1.95	88.9
NP_Small-Sparse_	497	0.416	9.67	0.184	2.45	80.1
^14^C-TAM						
NP_Large-Dense_	441	0.570	11.7	0.179	2.89	61.8
NP_Small-Sparse_	362	0.343	8.35	0.262	3.06	55.6

* Calculated using the trapezoidal method. The mean values of the three mice with normalized plasma concentrations were used. **: Calculated by the two-compartment model. The mean values of the three mice with normalized plasma concentrations were used.

## Data Availability

Data are contained within the article or Appendix A. The data presented in this study are available in inset articles or the Appendix A here.

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
