# Peer review of "Processing Parameters and Ion Excipients Affect the Physicochemical Characteristics of the Stereocomplex-Formed Polylactide-b-Polyethylene Glycol Nanoparticles and Their Pharmacokinetics"

_pharmaceutics, 2022, doi:10.3390/pharmaceutics14030568_

Round 1

Reviewer 1 Report

The title is too long and confusing, please shorten to 2-3 lines maximum and simplify.

The resolution of Fig. 3 needs to be improvement. Also the text needs revision as there are a lot of extra spaces.

Data Availability Statement is not filled, it still has the default text.

It would be interesting to see some microscopy images of the nmanoparticles.

Apart from that, the paper is fine and can be accepted for publication.

Author Response

First of all, we would like to express our appreciation to the reviewers for raising important issues and giving us helpful suggestions. We have revised our manuscript in the light of the reviewers’ comments. Our replies to each of the reviewer’s inquiries and the revised points are as follows.

  1. The title is too long and confusing, please shorten to 2-3 lines maximum and simplify.

A> We appreciate your thoughtful comments. We changed the title as follows.

“Processing parameters of anti-solvent precipitation procedures and ion excipients affect the physicochemical characteristics of the stereocomplex-formed polylactide-b-polyethylene glycol nanoparticles and their pharmacokinetics and biodistributions”

was changed to

“Processing parameters and ion excipients affect the physicochemical characteristics of the stereocomplex-formed polylactide-b-polyethylene glycol nanoparticles and their pharmacokinetics”.

  1. The resolution of Fig. 3 needs to be improvement. Also the text needs revision as there are a lot of extra spaces.

A> We appreciate your thoughtful comments. We substituted the Figure 3 to the high resolution one. Extra space was also deleted in the manuscript.

  1. Data Availability Statement is not filled, it still has the default text.

A> We are sorry for missing filling this form. Data Availability Statement was filled as follows.

“Data is contained within the article or supplementary material. The data presented in this study are available in insert articles or supplementary material here.”

  1. It would be interesting to see some microscopy images of the nmanoparticles.

A> We appreciate your thoughtful comments. Following the reviewer’s comments, we performed the field emission scanning electron microscopy for observations of the morphology of prepared nanoparticles. Based on the additional experiments, we revised the manuscript as follows.

  • We added the following methods for evaluations of morphology of nanoparticles. (Page 5, paragraph2)

“2.6.4 Evaluation of morphology of lyophilized NPs 

Morphology of lyophilized NPs were evaluated with field emission scanning electron microscopy (FE-SEM) according to a previously published method with slight modifications[27]. Lyophilized NPs were sputter-coated a layer of osmium with Neoc osmium coater. (Meiwafosis, Japan) Prepared samples were observed with FE-SEM (JEOL JSM-IT500HR LV, Japan) and operated at 10 kV.”

In the revised manuscript, we newly cited the following reference.

  1. Ghasemi, R., Abdollahi, M., Emamgholi Zadeh, E., Khodabakhshi, K., Badeli, A., Bagheri, H., & Hosseinkhani, S. (2018). MPEG-PLA and PLA-PEG-PLA nanoparticles as new carriers for delivery of recombinant human Growth Hormone (rhGH). Scientific Reports, 8(1), 1–13. https://doi.org/10.1038/s41598-018-28092-8
  • We added the figure 6D and its legend. (Page 11, paragraph 1)
  • We added the following results about evaluation of NPs morphology. (Page 10, paragraph 1)

“Morphology of prepared NPs were spherical. (Figure 5D) The lower panel shows morphology of NPs prepared with 100 mM LiBr/ TFR 3 mL/min, which was slightly larger than that of NPs prepared with 20 mM LiBr/ TFR 8 mL/min, (the upper panel).”

Reviewer 2 Report

Manuscript number: pharmaceutics-1585728

Title: Processing parameters of anti-solvent precipitation procedures and ion excipients affect the physicochemical characteristics of the stereocomplex-formed polylactide-b-polyethylene glycol nanoparticles and their pharmacokinetics and biodistributions

This manuscript describes the influence of a lithium bromide (as good solvent containing ion excipients) on the physicochemical properties of SC-PEG nanoparticles, as well as obtained using different processing parameters. The manuscript is interesting and good written, please see a few comments below:

# Page 1, lines 26-29, INTRODUCTION: Authors wrote: “Polymeric nanoparticles (NPs) have been intensively developed worldwide for various purposes, especially treatment of cancer [1-4]. Polymeric NPs have been chemically synthesized and modified to provide various physicochemical characteristics, such as hydrophilic-lipophilic balance, targeting moiety, size, and surface properties [5-7]. Because of the stealth properties of polyethylene glycol (PEG), polymeric NPs for passive targeting increase blood circulation time and accumulate in tumor tissues (i.e., enhanced permeability and retention effects) [8,9].” In addition, surface grafting can also be applied to introduce smart functionalities for a broad spectrum of applications. For illustration, This strategy enables the fabrication of highly functional biopolymer-based scaffolds for tissue engineering or can be applied to build hybrid complex systems for controlled-release delivery of chemotherapeutics. Please add some information (other perceptiveness strategies) about these possibilities in this paragraph based on new references, please see for example: doi.org/10.1016/B978-0-12-810462-0.00004-1

# Pages 14, lines 518-534, CONCLUSIONS: The authors described more important conclusions from the obtained research results, but please add in this section more information about the application aspects of the obtained experimental results and the novelty of your research.

I recommend the paper for publication in Pharmaceutics but after minor revision.

Author Response

First of all, we would like to express our appreciation to the reviewers for raising important issues and giving us helpful suggestions. We have revised our manuscript in the light of the reviewers’ comments. Our replies to each of the reviewer’s inquiries and the revised points are as follows

  1. # Page 1, lines 26-29, INTRODUCTION: Authors wrote: “Polymeric nanoparticles (NPs) have been intensively developed worldwide for various purposes, especially treatment of cancer [1-4]. Polymeric NPs have been chemically synthesized and modified to provide various physicochemical characteristics, such as hydrophilic-lipophilic balance, targeting moiety, size, and surface properties [5-7]. Because of the stealth properties of polyethylene glycol (PEG), polymeric NPs for passive targeting increase blood circulation time and accumulate in tumor tissues (i.e., enhanced permeability and retention effects) [8,9].” In addition, surface grafting can also be applied to introduce smart functionalities for a broad spectrum of applications. For illustration, This strategy enables the fabrication of highly functional biopolymer-based scaffolds for tissue engineering or can be applied to build hybrid complex systems for controlled-release delivery of chemotherapeutics. Please add some information (other perceptiveness strategies) about these possibilities in this paragraph based on new references, please see for example: doi.org/10.1016/B978-0-12-810462-0.00004-1

A> We appreciate your thoughtful comments. Following the reviewer’s comments, we newly added the sentences as follows in the text.

“In addition, surface modified, or grafted NPs were investigated for active targeting or tissue engineering. [10,11] Thus polymeric NPs would be promising delivery systems.”

In addition, we newly cited the following references in the text. (Page 1, 1 paragraph).

  1. Bertranda N., Wu J., Xu X., Kamaly N., Farokhzad O. C., Cancer nanotechnology: The impact of passive and active targeting in the era of modern cancer biology (2014), Adv. Drug Deliv. Rev., 66, 2-25, https://doi.org/10.1016/j.addr.2013.11.009
  2. KyzioÅ‚ A. and KyzioÅ‚ K., Surface Functionalization With Biopolymers via Plasma-Assisted Surface Grafting and Plasma-Induced Graft Polymerization—Materials for Biomedical Applications, Biopolymer Grafting, 1st ed.; Thakur V. K. Eds; Elsevier, Amsterdam, Nederland, 2018; pp.115-151, https://doi.org/10.1016/B978-0-12-810462-0.00004-1

  1. # Pages 14, lines 518-534, CONCLUSIONS: The authors described more important conclusions from the obtained research results, but please add in this section more information about the application aspects of the obtained experimental results and the novelty of your research.

A> We appreciate your thoughtful comments. Following the reviewer’s comments, we revised the sentences as follows. (Page 16, 1 paragraph).

“~ Processing parameters altered the PK profiles and tissue distributions. “

was changed to

“~Processing parameters altered the PK profiles and tissue distributions. These findings should be helpful to make the manufacturing process of polymeric NPs robust and to control their critical quality attributes.

Reviewer 3 Report

The manuscript entitled "Processing parameters of anti-solvent precipitation procedures and ion excipients affect the physicochemical characteristics of the stereocomplex-formed polylactide-b-polyethylene glycol nanoparticles and their pharmacokinetics and biodistributions" reports the development of SC-PEG NPs via continuous anti-solvent precipitation with a T-junction method and investigated the effect of ion excipient (LiBr)-based solvents on the characteristics of SC-PEG NPs by varying the processing temperature and total flow rate (TFR); and demonstrated the properties such as Dh, surface PEG densities, and DLC, which altered the PK profiles and tissue distributions. The objective of the study is good and the obtained results are interesting. However, the manuscript lacks in critical analysis and discussion of the data obtained and validation of the claims. Therefore, the manuscript requires a moderate revision before its publication in the journal.

  1. Title should be revised
  2. What is the size of nanoparticles obtained? Please provide appropriate experimental data for the same (preferably electron microscopy/AFM images)
  3. The schematic representation of the developed NP system and their functional interaction with the properties/target-application should be provided
  4. The mechanism of LiBr in property enhancements and in the performed applications should be discussed
  5. How the concentration of LiBr influences the properties? Mechanism should be discussed
  6. Provide the optical microscopy images of the specimen and data discussed in Fig. 6
  7. Authors have presented the observations and no discussion on the mechanism or reasons behind the property changes due to the materials and their physiochemical conditions have been described
  8. English of the manuscript should be improved

Author Response

First of all, we would like to express our appreciation to the reviewers for raising important issues and giving us helpful suggestions. We have revised our manuscript in the light of the reviewers’ comments. Our replies to each of the reviewer’s inquiries and the revised points are as follows.

  1. What is the size of nanoparticles obtained? Please provide appropriate experimental data for the same (preferably electron microscopy/AFM images)

A> We appreciate your thoughtful comments. Following the reviewer’s comments, we performed the field emission scanning electron microscopy for observations of the morphology of prepared nanoparticles. Based on the additional experiments, we revised the manuscript as follows.

  • We add the following methods for evaluations of morphology of nanoparticles. (Page 5, paragraph2)

“2.6.4 Evaluation of morphology of lyophilized NPs 

Morphology of lyophilized NPs were evaluated with field emission scanning electron microscopy (FE-SEM) according to a previously published method with slight modifications[27]. Lyophilized NPs were sputter-coated a layer of osmium with Neoc osmium coater. (Meiwafosis, Japan) Prepared samples were observed with FE-SEM (JEOL JSM-IT500HR LV, Japan) and operated at 10 kV.”

In the revised manuscript, we newly cited the following reference.

  1. Ghasemi, R., Abdollahi, M., Emamgholi Zadeh, E., Khodabakhshi, K., Badeli, A., Bagheri, H., & Hosseinkhani, S. (2018). MPEG-PLA and PLA-PEG-PLA nanoparticles as new carriers for delivery of recombinant human Growth Hormone (rhGH). Scientific Reports, 8(1), 1–13. https://doi.org/10.1038/s41598-018-28092-8
  • We add the figure 6D and its legend. (Page 11, paragraph 1)
  • We add the following results about evaluation of morphology. (Page 10, paragraph 1)

“Morphology of prepared NPs were spherical. (Figure 5D) The lower panel shows morphology of NPs prepared with 100 mM LiBr/ TFR 3 mL/min, which was slightly larger than that of NPs prepared with 20 mM LiBr/ TFR 8 mL/min, (the upper panel).”

  1. The schematic representation of the developed NP system and their functional interaction with the properties/target-application should be provided

A> We appreciate your thoughtful comments. Following the reviewer’s comments, we revised the sentences as follows.

The schematic representation of nanoparticle preparation was added as a Figure 1. (Page 4)

The schematic brief summary of this study was added as a Figure 8. (Page 16)

  1. The mechanism of LiBr in property enhancements and in the performed applications should be discussed

A> We appreciate your thoughtful comments. Following the reviewer’s comments, we newly added the sentences as follows. (Page 14, 1 paragraph)

“In contrast, the addition of over 100 mM LiBr would lead salting-out effect, which re-sulted in associating the unimers with their hydrophobic domains. These would contribute the increase of Nass in the formation of NPs. Previous research showed that increment of LiBr in the DMF increased polymer-polymer interactions [36]. It was also reported that size of lignin nanoparticles prepared by anti-solvent precipi-tations was controlled by structures of lignin in the solvent. The solution structures of lignin were changed by ion strength or pH [37]. Our observations were consistent with these reports.”

In addition, we newly cited the following reference in the text.

  1. Zhao W., Simmons B., Singh S., Ragauskas A., and Cheng G. (2016), From lignin association to nano-/micro-particle preparation: extracting higher value of lignin, Green Chem., 18, 5693-5700, 10.1039/C6GC01813K

  1. How the concentration of LiBr influences the properties? Mechanism should be discussed

A> We appreciate your thoughtful comments. Following the reviewer’s comments, we newly added the sentences as follows. (Page 13 2 paragraph)

“It was speculated that the addition of LiBr up to 20 mM resulted in dissociating the interactions between hydrophilic domain of unimer because previous research eluci-dated that LiBr shield the dipole moment between polymers. [35] In contrast, the addi-tion of over 100 mM LiBr would lead salting-out effect, which resulted in associating the unimers with their hydrophobic domains. These would contribute the increase of Nass in the formation of NPs. Previous research showed that increment of LiBr in the DMF in-creased polymer-polymer interactions [36].

In addition, we newly cited the following references in the text.

  1. Hann N.D. (1977), Effects of lithium bromide on the gel-permeation chromatography of polyester-based polyurethanes in dimethylformamide, J. Polym. Sci., Part A: Gen. Pap., 15(6), 1331-1339, https://doi.org/10.1002/pol.1977.170150604
  2. Idris A. and Ahmed I. (2008), Viscosity behavior of microwave-heated and conventionally heated poly(ether sulfone)/dimethylformamide/lithium bromide polymer solutions, J. Appl. Polym. Sci., 108(1), 302-307, https://doi.org/10.1002/app.27590

  1. Provide the optical microscopy images of the specimen and data discussed in Fig. 6

A> We appreciate your thoughtful comments. We understand that cellular population incorporated the NP in the organ was important for elucidating the distribution mechanism. However, the scope of this present study was elucidating the relationship between formulation process parameter, formulation characteristics and pharmacokinetics to clarify the critical process parameters. Thus, optical microscopy images could be the out of scope in this study. We are now considering the idea of investigating this point in a future paper.

  1. Authors have presented the observations and no discussion on the mechanism or reasons behind the property changes due to the materials and their physiochemical conditions have been described

A> We appreciate your thoughtful comments. Following the reviewer’s comments, we added the sentences as follows in the text. (Page 14 1 paragraph)

“Figure 8 shows the brief summary of the relationship between process parameters in the preparation of NPs. The addition of LiBr in the good solvent caused association of polymer and slower flow rate decreased diffusion rate of good solvent, which led to increasing Nass. Dh was increased by increasing Nass and the specific surface area of NPs were decreased, which resulted in increase of surface PEG densities. These were the reasons why processing parameters affect the nanoparticle characteristics.”

  1. English of the manuscript should be improved

A> English proofreading was conducted by Editage (www.editage.jp/).

Round 2

Reviewer 3 Report

Authors have revised the manuscript satisfactorily and it can be accepted for the publication.